# Vaccine Efficacy of a Replication-Competent Interferon-Expressing Porcine Reproductive and Respiratory Syndrome (PRRS) Virus Against NADC-34 Challenge

**DOI:** 10.3390/vaccines13040413

**Published:** 2025-04-15

**Authors:** Laura C. Miller, Sarah J. Anderson, Alexandra C. Buckley, Erin E. Schirtzinger, Mahamudul Hasan, Kaitlyn M. Sarlo Davila, Damarius S. Fleming, Kelly M. Lager, Jiuyi Li, Yongming Sang

**Affiliations:** 1Diagnostic Medicine and Pathobiology, College of Veterinary Medicine, Kansas State University, 1800 Denison Ave, Manhattan, KS 66506, USA; eesem@vet.k-state.edu (E.E.S.); mahamudul@vet.k-state.edu (M.H.); 2Virus and Prion Research Unit, National Animal Disease Center, United States Department of Agriculture-Agricultural Research Service, 1920 Dayton Ave, Ames, IA 50010, USA; sarah.j.anderson@usda.gov (S.J.A.); alexandra.buckley@usda.gov (A.C.B.); kelly.lager@usda.gov (K.M.L.); 3Ruminant Diseases and Research Unit, National Animal Disease Center, United States Department of Agriculture-Agricultural Research Service, 1920 Dayton Ave, Ames, IA 50010, USA; kaitlyn.sarlodavila@usda.gov; 4Animal Parasitic Diseases Laboratory, Beltsville Agricultural Research Center, United States Department of Agriculture-Agricultural Research Service, 10300 Baltimore Ave, Beltsville, MD 20705, USA; damarius.fleming@usda.gov; 5Department of Food and Animal Sciences, College of Agriculture, 3500 John A. Merritt Blvd, Tennessee State University, Nashville, TN 37209, USA; jli4@tnstate.edu

**Keywords:** porcine reproductive and respiratory syndrome virus, antiviral, Type I interferons, modified live vaccine, reverse genetics, viral vaccine expression vector, pig viral infections

## Abstract

**Background/Objectives**: Porcine reproductive and respiratory syndrome virus (PRRSV) significantly impedes swine production due to rapid genetic variation and suppression of antiviral interferon (IFN) responses, leading to ineffective immunity. To address this, we developed IFNmix, a replication-competent PRRSV modified live vaccine (MLV) candidate co-expressing three Type I IFN subclasses (IFNα, IFNβ, IFNδ) to enhance antiviral immunity. **Methods**: In two independent in vivo experiments, we compared the protection of IFNmix and a commercial PRRSV MLV vaccine during challenge with a virulent PRRSV strain. Clinical signs, antibody and cytokine production, viral replication, and lung pathology in IFNmix-vaccinated pigs were compared to those of commercial PRRSV vaccines and controls. **Results**: Pigs vaccinated with IFNmix exhibited similar anti-PRRSV antibody development, serum viral loads, lung lesions, and cytokine responses post-challenge with the virulent NADC34 strain, with comparable or lower body temperatures and weight gain, to pigs vaccinated with the commercial vaccines. While IFNmix showed early viral load reduction compared to the commercial vaccine (Days 7–14 post-challenge), it demonstrated similar efficacy in controlling PRRSV replication and lung pathology. **Conclusions:** These findings suggest that IFNmix, by expressing multiple IFNs, can potentially enhance innate and adaptive immune responses, offering a promising approach to improving PRRSV vaccine efficacy. Further studies are needed to evaluate IFNmix against a broader range of PRRSV strains and to optimize its attenuation and immunogenicity.

## 1. Introduction

Since the emergence of porcine reproductive and respiratory syndrome virus (PRRSV) in the late 1980s, the prevention of PRRSV infections and associated diseases has been a top priority for swine producers worldwide. PRRSV is responsible for over $1 billion in losses annually to the US swine industry [1,2,3]. PRRSV (order *Nidovirales*, family *Arteriviridae*) is an enveloped virus with a single-stranded, positive-sense RNA genome of approximately 15 kb that encodes eight structural and fifteen non-structural proteins (nsps). Control of PRRSV is especially challenging due to the high degree of genetic diversity among PRRSV strains (high viral mutation and recombination rates) which limits the efficacy of vaccine-induced immunity against heterologous strains [4,5,6,7,8]. Additionally, several PRRSV non-structural proteins (nsp1α, nsp1β, nsp2, nsp4, nsp11) and the nucleoprotein, N, have been shown to suppress Type I interferon induction and signaling pathways through multiple mechanisms [9,10,11]. Type I interferons (IFNs) work in an autocrine and paracrine manner, inducing the expression of interferon-stimulated genes that create an antiviral state in infected and neighboring cells, while also initiating adaptive immunity including humoral and cell-mediated responses [12]. PRRSV suppression of Type I IFNs results in an ineffective innate immune response and a weak, delayed adaptive immune response that lacks immune memory [6]. This delay in the development of neutralizing antibodies required for viral clearance may contribute to the persistence of viral replication and shedding [11,13,14], as well as the pathology associated with some PRRSV strains.

Despite the consensus that PRRSV actively suppresses IFN production and signaling, the extent to which IFN is induced varies between strains [8,15,16]. Nan et al. [16] found the A2MC2 strain induced significantly more IFN during infection of MARC-145 and PAM cells compared to PRRSV VR-2332 and the Ingelvac modified live vaccine (MLV) which share 99.8% nucleotide identity. Additionally, A2MC2 showed no cytopathic effect during infection. Later, Wang et al. [8] compared neutralizing antibody development during in vivo infection with A2MC2, the Ingelvac MLV vaccine, and PRRSV VR-2385. Pigs infected with A2MC2 developed detectable neutralizing antibodies earlier (28 days post-infection (dpi) vs. 35 and 56 dpi, respectively). Despite this, A2MC2 infection produced lung lesions of similar severity to the virulent strain VR-2385 at 14 dpi. These and other findings have led some researchers to suggest that suppression of Type I IFNs could be overcome by the addition of IFNs to a PRRSV MLV vaccine leading to a more robust humoral and cell-mediated adaptive response [17,18,19,20].

Pigs have a large Type I IFN complex, including 60 functional IFN genes in three multi-gene [alpha (α), delta (δ), and omega (ω)] and three single gene [beta (β), epsilon (ε) and kappa (κ)] subclasses [21,22]. Not all IFN subclasses are produced in all organs/cell types, nor do they all induce similar antiviral states [18,23,24,25,26,27]. When MARC-145 cells or PAMs were treated with each subclass or each subtype within the multi-gene subclasses, a range of antiviral activity against PRRSV was seen. Subtypes of IFNα (except 7 and 11) rendered MARC-145 cells and PAMs resistant to PRRSV infection. However, treatment with IFNβ, IFNδ (except 2 and 7), or IFNω also protected PAMs from PRRSV [18].

The current study describes the preliminary evaluation of a replication-competent PRRSV MLV candidate (IFNmix) engineered to co-express three Type I IFN subclasses (IFNα, IFNβ, IFNδ) in vivo. Two experiments were conducted to compare the IFNmix MLV candidate to two different PRRSV MLV vaccines. Overall, vaccination with the IFNmix MLV candidate showed similar anti-PRRSV antibody development to both commercial vaccines. Following challenge, IFNmix showed similar serum viral loads, lung lesions, and cytokine responses to the two commercial vaccines.

## 2. Materials and Methods

### 2.1. Replication-Competent Recombinant Porcine Reproductive and Respiratory Syndrome (PRRS) Viruses Expressing Antiviral Cytokines Interferonα, Interferonβ, and Interferonδ

The infectious cDNA clone, pCMV-S-P129, was provided by Zoetis and the expression vector, pCMV-S-P129-1bGFP2, was constructed as previously described [28]. The open reading frames of antiviral IFNα, IFNβ, and IFNδ were inserted between the *Afl* II and *Mlu* I restriction sites at the viral ORF1b/ORF2 junction region. The expressed IFNα, IFNβ, and IFNδ peptides each contained a His-tag containing a C-terminal protease cleavage site, which allowed for the detection of replication-competent IFN expression, as well as the detection of conditional IFN activation [29]. Two μg of plasmid DNA was transfected into MARC-145 cells in 24-well plates using Lipofectamine™ 2000 (Invitrogen). Subsequent virus yields were measured by end-point titration of culture media on MARC-145 cells and PAMs. Culture supernatants from cells transfected with infectious clones were harvested at 5 days post-transfection and designated ‘passage (P) 1’. The P1 virus was used to inoculate fresh MARC-145 cells to collect P2 then P3 at an interval of 4–5 days between successive passages. Each passage virus was titrated, aliquoted, and stored at −80 °C until use.

### 2.2. Animal Study Design

IFNmix MLV was evaluated in two independent experiments. In both experiments, three-week-old pigs were purchased from a commercial breeder and randomly assigned to four treatment groups (N = 10): Sham vaccine/Sham challenge (Sham/Sham), Sham vaccine/PRRSV challenge (Sham/PRRSV), IFNmix MLV vaccine/PRRSV challenge (IFNmix/PRRSV), commercial MLV vaccine/PRRSV challenge (ComMLV/PRRSV). The two experiments evaluated IFNmix MLV against a different commercial MLV vaccine (ComMLV1, ComMLV2). Groups were individually housed in ABSL-2 animal rooms. Food and water were provided ad libitum. Forty-two days prior to challenge, groups (N = 10/group) were intramuscularly vaccinated with either 2 mL of 1 × 10^4^ TCID_50_/mL of the IFNmix MLV, 2 mL of 1 × 10^4^ TCID_50_/mL of one of two commercial PRRSV MLVs, or 2 mL of cell culture media. On the day of challenge, PRRSV challenge groups received 2 mL of 1 × 10^4^ TCID_50_/mL of PRRSV NADC34 intramuscularly [30]. The Sham challenge group received 2 mL of cell culture media.

### 2.3. Clinical Signs and Sampling

During both experiments, pigs were checked for signs of illness daily and body temperatures were recorded daily using a microchip, (Destron Fearing, Dallas, TX, USA) placed in the neck. Body weights were recorded weekly prior to and after challenge in both experiments. Serum was separated by centrifugation of blood samples collected weekly and stored at −80 °C until assayed. At 14 dpc, pigs were humanely euthanized and necropsied.

### 2.4. ELISA

The IDEXX PRRS X3 Ab test ELISA (IDEXX Laboratories, Westbrook, ME) was used to quantify serum anti-PRRSV antibodies at Days –42, 0, and 14 dpc for both experiments. In addition, ELISAs were conducted on serum samples from Days –35, –28, –21, –14, and –7 of Experiment 2 to track the development of anti-PRRSV antibodies during vaccination. Assays were conducted according to manufacturer’s instructions. Samples with S/P ratios greater or equal to 0.4 are reported as positive.

### 2.5. PRRSV RT-qPCR

PRRSV RT-qPCR was performed on serum from Days –42, 0, 7, and 14 dpc in both experiments as previously described [30]. Additionally, to track viral replication post-vaccination, RT-qPCR was also conducted on serum samples from Days –35, –28, –21, –14, –7 in Experiment 1. Briefly, viral RNA in serum samples was extracted using the MagMax-96 Viral RNA kit (Applied Biosystems, Waltham, MA, USA) following the manufacturers’ recommendations on an Applied Biosystems MagMax Express Magnetic Particle Processor. The AgPath ID One Step RT-qPCR kit (Applied Biosystems) was used with primers and probe targeting ORF-7 of the PRRSV genome. Assays were run on an Applied Biosystems 7500 Real-Time PCR System in duplicate. Viral RNA copies were quantified by including a serial dilution of transcribed RNA in each plate. Results are reported as log_10_ RNA copies.

### 2.6. Cytokine Assays

Serum cytokine concentrations were measured on 0 and 7 dpc in both experiments using the ProcartaPlex Porcine Cytokine 9-Plex Assay (Invitrogen) according to manufacturer’s instructions with samples diluted 1:2 and two technical replicates. Cytokines were measured on a Luminex MAGPIX instrument. Due to many cytokine median fluorescence intensities (MFIs) being below the lowest standard protein concentration, MFI was analyzed [31]. The median fluorescence value for each sample was log_10_ transformed prior to statistical analysis.

### 2.7. Lung Lesions

Following necropsy at 14 dpc, lungs were inspected for macroscopic lesions and the percentage of affected lung surface area was recorded for each lung lobe [32]. The average lesion percentage for each lobe is reported for each group.

### 2.8. Statistical Analyses

Statistical analyses were conducted in GraphPad Prism 10 (Dotmatics). To determine if the two experiments could be combined, significant differences between treatment groups were identified using a one-way ANOVA (α = 0.05) with appropriate multiple comparison tests. Simple linear regression was used to plot a best-fit line for weight gain data by treatment group. Slopes of the best-fit lines were then tested for statistical significance. A two-way ANOVA, α = 0.05, with time and treatment as factors, was used analyze the results of the bead-based multiplex cytokine assays. Radar charts for macroscopic scoring of lung lesions were created in RStudio version 4.4.1 using the fmsb and scales libraries in the ggplot2 package.

## 3. Results

Results from the two experiments were first compared for significant differences to determine if results could be combined (Appendix A). In all analyses, there was at least one case where results were significantly different between the two experiments. Therefore, the results of each experiment are reported, and generalizations are made where possible. Details of statistical analyses mentioned in the text are found in Appendix A.

### 3.1. Temperature

Body temperatures were similar between the treatment groups from Day –42 to Day 0 in both experiments (Appendix A) but significant differences were seen after challenge (Appendix A). In Experiment 1, the ComMLV1/PRRSV group had the highest average body temperature (above 40.5 °C) from 1 day post-challenge (dpc) to 5 dpc (Figure 1a). The IFNmix/PRRSV group’s average body temperature was also elevated from 1 dpc until its peak at 5 dpc. After 5 dpc, both the ComMLV1/PRRSV group’s and the IFNmix/PRRSV group’s average body temperatures decreased in a linear fashion until 13 dpc and 11 dpc, respectively, when average body temperatures returned to Day 0 values. The Sham/PRRSV group’s average body temperature was elevated to approximately 40 °C until 4 dpc. Their temperature peaked at 6 dpc at approximately 41 °C and then remained near 41 °C until 12 dpc. In Experiment 2, the IFNmix/PRRSV and the Sham/PRRSV groups follow a similar pattern to Experiment 1 (Figure 1b). The IFNmix/PRRSV group’s average body temperature peaked at 5 dpc and then decreased linearly until 12 dpc; while the average body temperature of the Sham/PRRSV group did not become elevated until 4 dpc and then peaked at 5 dpc and again at 9 dpc. The ComMLV2/PRRSV used in Experiment 2 had a body temperature pattern similar to IFNmix/PRRSV except that it dropped to Day 0 levels at 8 dpc. In both experiments, the IFNmix/PRRSV groups had body temperatures similar to or lower than temperatures seen in the ComMLV/PRRSV groups.

### 3.2. Weight

A simple linear regression was used to create a line describing the increase in weight of each treatment group throughout the experiment. Equations describing the line fit to the data points for each treatment are listed in Appendix A. The slopes of the lines for each treatment were first compared between Experiment 1 and Experiment 2 (Appendix A). Sham/PRRSV, Sham/Sham, and ComMLV/PRRSV showed no statistically significant differences between the slope of the line in Experiment 1 and Experiment 2 (*p* = 0.6356, *p* = 0.7433, *p* = 0.3372, respectively). In contrast, the slopes of the lines for the IFNmix/PRRSV group were significantly different (*p* < 0.0001) between Experiment 1 and Experiment 2. Therefore, the data could not be combined (Figure 2). In Experiment 1, the slope of the Sham/Sham line was significantly greater than IFNmix/PRRSV, ComMLV1/PRRSV, and Sham/PRRSV (*p* < 0.0001, *p* = 0.0080, *p* < 0.0001, respectively) as expected (Figure 2a). The slope of the ComMLV1/PRRSV line was also significantly greater than the slopes of IFNmix/PRRSV and Sham/PRRSV (*p* < 0.0001, *p* = 0.0040, respectively). The slope of the IFNmix/PRRSV line was significantly lower than Sham/PRRSV (*p* = 0.0098). This was in contrast to Experiment 2, where the slope of IFNmix/PRRSV was not significantly different from ComMLV2/PRRSV (*p* = 0.4983) (Figure 2b). In Experiment 2, ComMLV2/PRRSV was not significantly different than Sham/Sham (*p* = 0.0519) but was significantly greater than Sham/PRRSV. Additionally, in Experiment 2, the Sham/PRRSV line had the smallest slope.

### 3.3. Serum Antibodies

All pigs were negative for anti-PRRSV antibodies at Day –42 in both experiments (Figure 3a,b). On Day 0, the IFNmix- and ComMLV-vaccinated pigs were positive for anti-PRRSV antibodies, showing successful seroconversion. In both experiments, no statistically significant difference was seen in serum antibody concentrations between IFNmix- and ComMLV1- or ComMLV2-vaccinated pigs (*p* = 0.1468, *p* = 0.2352, respectively). By 14 dpc, Sham/PRRSV, IFNmix/PRRSV, and ComMLV/PRRSV groups were all positive for anti-PRRSV antibodies. In Experiment 1, there was no significant difference in antibody levels between Sham/PRRSV and IFNmix/PRRSV (*p* = 0.7255) and between IFNmix/PRRSV and ComMLV1/PRRSV (*p* = 0.2269) (Figure 3a). However, the serum antibody levels were significantly different between Sham/PRRSV and ComMLV1/PRRSV (*p* = 0.0182). In contrast, Experiment 2 serum antibody levels were significantly different between ComMLV2/PRRSV and both Sham/PRRSV (*p* < 0.0001) and IFNmix/PRRSV (*p* = 0.0003) (Figure 3b). The data from both experiments show that IFNmix induced similar anti-PRRSV antibody levels to commercially available modified live vaccines during PRRSV infection.

In Experiment 2, serum anti-PRRSV antibody levels were evaluated post-vaccination, as well as after challenge (Figure 3c). All IFNmix- or ComMLV2-vaccinated pigs had seroconverted within 14 days of vaccination. This also marks the peak anti-PRRSV antibody level for IFNmix-vaccinated pigs, which remained steady until 14 dpc. In contrast, the ComMLV2-vaccinated pigs’ serum antibody levels peaked 35 days after vaccination and remained similar until Day 0. The serum antibody levels induced by ComMLV2 vaccination are significantly greater than IFNmix at 28 and 35 days after vaccination (*p* = 0.0334, *p* = 0.0212, respectively).

### 3.4. Viral Load

The two experiments did not show any statistically significant differences when the log_10_ PRRSV ORF-7 RNA copy numbers at each time point were compared (Appendix A). Additionally, the differences in log_10_ RNA copy numbers between time points within treatments showed similar results between experiments (Appendix A). Figure 4 shows the log_10_ RNA copy number of each treatment at the indicated time points in both experiments. The experiments showed a similar pattern of significance between treatments at each time point. Small amounts of PRRSV RNA were detectable at Day 0 in the IFNmix/PRRSV and ComMLV/PRRSV groups, which was not unexpected due to the vaccines being replication competent (Figure 4a,b). At 7 dpc, Sham/PRRSV, IFNmix/PRRSV, and ComMLV1/PRRSV treatments had significantly more RNA copies than Sham/Sham (*p* < 0.0001, *p* = 0.0018, respectively). The Sham/PRRSV group had the highest log_10_ RNA copy number at 7 dpc, which was not significantly different from ComMLV(1 or 2)/PRRSV (*p* = 0.1584, *p* = 0.2562, respectively), but it was significantly higher than IFNmix/PRRSV (*p* < 0.0001, *p* = 0.0026, respectively). IFNmix/PRRSV, on the other hand, was significantly different from ComMLV1/PRRSV (*p* = 0.0035) but similar to ComMLV2/PRRSV (*p* = 0.3069). At 14 dpc, Sham/PRRSV also had the highest log_10_ RNA copy number (Figure 4a,b). In both experiments, RNA copy numbers for both IFNmix/PRRSV and ComMLV(1 or 2)/PRRSV were significantly reduced compared to Sham/PRRSV (*p* < 0.0001, *p* < 0.0001, respectively). These results showed that the IFNmix vaccine reduced replication of PRRSV better (but not significantly) than two commercial MLV vaccines after challenge.

In Experiment 1, replication of the IFNmix MLV and ComMLV1 was monitored weekly by RT-qPCR until challenge at Day 0 (Figure 4c). Seven days post-vaccination (dpv), replication of IFNmix MLV was significantly higher than ComMLV1 (*p* = 0.0009), but at all later time points there was no significant difference between IFNmix MLV and ComMLV1. The mean IFNmix/PRRSV RNA copy number remained steady at ~3.5 log_10_ from 7 to 21 dpv with individual samples ranging from ~2.0 to 5.5 log_10_ RNA copies. The mean ComMLV/PRRSV RNA copy number, during this same period, remained at or below 3.0 log_10_ copies with samples having a wide range of values. The mean RNA copy number for both groups began to decline at 28 dpv, and by challenge at Day 0, replication was essentially undetectable in most vaccinated pigs.

### 3.5. Serum Cytokines

Nine serum cytokine concentrations were determined for all treatment groups on 0 and 7 dpc (IFNα, IFNγ, IL-1b, IL-4, IL-6, IL-8, IL-10, IL-12p40, TNFα). Comparisons of mean cytokine concentrations between Experiment 1 and Experiment 2 showed many significant differences (Appendix A), with Experiment 2 having consistently lower levels in 7 of the 9 cytokines. IFNα and IL-12p40 showed no difference in concentration between experiments.

Figure 5 shows the comparisons between treatments at 0 and 7 dpc for each cytokine, where the cytokines have been grouped into Th1-associated (IFNα, IFNγ, IL-1b, IL-8, IL-12p40, TNFα) and Th2-associated (IL-4, IL-6, IL-10) responses. In Experiment 1 (Figure 5a), on Day 0 there were no significant differences between treatments for most Type 1 cytokines. However, IFNmix/PRRSV had a significantly higher IL-8 concentration than ComMLV1/PRRSV (*p* = 0.0159) and the TNFα concentration for IFNmix/PRRSV was significantly higher than Sham/PRRSV and Sham/Sham (*p* < 0.0001 and *p* = 0.0327, respectively). In Experiment 2 (Figure 5b), the only Type 1 cytokine that showed a statistically significant difference between treatments on Day 0 was IFNα, where IFNmix/PRRSV was significantly higher than Sham/Sham (*p* = 0.0107). This result, however, could have been a residual effect of the IFNmix MLV.

At 7 dpc, significant differences were seen between treatments for IFNα, IFNγ, IL-8, and TNFα in Experiment 1. Aside from IFNα, where Sham/PRRSV was significantly higher than the other treatments (*p* < 0.0001, *p* = 0.0023, *p* < 0.0001), the significant differences in IFNγ, IL-8, and TNFα were between IFNmix/PRRSV or ComMLV1/PRRSV and the positive (Sham/PRRSV) or negative (Sham/Sham) controls. No statistically significant differences were seen between IFNmix/PRRSV and ComMLV1/PRRSV for any cytokine at 7 dpc. Experiment 2 showed a similar pattern in which significant differences between treatments were seen in IFNα, IL-8, IL-12p40, and TNFα. For Type 2 (IL-4, IL-6, IL-10) cytokines (Figure 5c,d), significant differences were only seen in IL-10 at 7 dpc between IFNmix/PRRSV and ComMLV1/PRRSV and controls. No significant differences were seen between IFNmix/PRRSV and ComMLV1/PRRSV or ComMLV2/PRRSV for Type 2 cytokines.

Comparisons of cytokine levels within each treatment between Day 0 and 7 dpc are shown in Figure 6. In Experiment 1, IFNα levels were significantly higher at 7 dpc for Sham/PRRSV (*p* < 0.0001) and ComMLV1/PRRSV (*p* < 0.0001). Sham/PRRSV showed a significant decrease in IL-8 levels at 7 dpc (*p* < 0.0001). ComMLV1/PRRSV showed a significant increase in IL-12p40 levels at 7 dpc (*p* = 0.0056). Experiment 2 showed the same significant increases in IFNα for Sham/PRRSV (*p* < 0.0001) and ComMLV2/PRRSV (*p* = 0.0018). The significant increase in IL-12p40 seen in ComMLV1/PRRSV in Experiment 1 at 7 dpc was also seen in Experiment 2 in ComMLV2/PRRSV. IFNmix/PRRSV and ComMLV2/PRRSV both showed significantly increased TNFα levels at 7 dpc (*p* = 0.0003, *p* = 0.0092), in contrast to Experiment 1. Results for Type 2 cytokines showed some variation between the two experiments. At 7 dpc in Experiment 1, IFNmix/PRRSV showed a significant reduction in IL-4 (*p* = 0.0467), Sham/PRRSV showed a significant increase in IL-6 *p* = 0.0004), and ComMLV1/PRRSV showed a significant increase in IL-10 (*p* = 0.0036). In Experiment 2 at 7 dpc, ComMLV2/PRRSV showed significant increases in IL-4 (*p* = 0.0021) and IL-10 (*p* = 0.0006), Sham/PRRSV showed significant increases in IL-6 (*p* = 0.0080) and IL10 (*p* = 0.0487), and IFNmix/PRRSV showed a significant increase in IL-10 (*p* < 0.0001).

### 3.6. Lung Lesions

In Experiment 1, lung lesions were found in all lobes of both lungs in all treatment groups, ranging from approximately 10% in Sham/Sham to 30–50% in Sham/PRRSV, IFNmix/PRRSV, and ComMLV1/PRRSV (Figure 7a). Statistical analysis showed that for all lobes, the mean percentage affected was greater in Sham/PRRSV and IFNmix/PRRSV than Sham/Sham. Lung lesions in ComMLV1/PRRSV, however, were similar to Sham/Sham in the accessory and right cranial lobes but were statistically greater than Sham/Sham in the other lobes. The lung lesions in IFNmix/PRRSV and ComMLV1/PRRSV were not statistically different (Figure 7b).

In Experiment 2, a different pattern of lesions was observed in which the caudal lobes were less affected than the other lobes (Figure 7c). Sham/PRRSV had the greatest mean percentage of affected lung tissue in all lobes (~60%, ~15% for caudal lobes) with a mean percentage of affected lung significantly greater than all other treatment groups in all lobes except caudal lobes. IFNmix/PRRSV had significantly more affected lung tissue than Sham/Sham in the left upper lobe, the accessory lobe, and the right middle lobe. ComMLV2/PRRSV had significantly more affected lung tissue than Sham/Sham in the left middle lobe, the accessory lobe, and the right middle lobe. However, IFNmix/PRRSV and ComMLV2/PRRSV did not show significant differences in lung lesion in any lobe (Figure 7d).

## 4. Discussion

Control of PRRSV through vaccination has been especially challenging due to the virus’ ability to suppress interferon production and signaling which is necessary to initiate and sustain a robust humoral and cell-mediated immune response and memory to vaccination. Additionally, PRRSV strains are genetically diverse due to high rates of mutation and recombination, thus increasing the difficulty of generating vaccine protection against heterologous strains. Commercial PRRSV MLV vaccines have been in use for over 20 years with limited efficacy as shown by the continued circulation of genetically similar strains [6,15]. In general, MLVs are preferred over inactivated vaccines because they induce a more robust immune response. However, in the case of PRRSV MLV vaccines, the presence of non-structural proteins that actively suppress Type I IFN production and immune signaling may be the cause of reduced vaccine response and protection.

Improvements in reverse genetics, gene-editing, vaccine vectors, and sequencing technologies (i.e., spatial transcriptomics) now provide researchers with powerful tools to decipher the intricacies of virus–host immune interactions and could lead to improved vaccines. Using reverse genetics, we have engineered recombinant PRRS viruses that express the antiviral cytokine IFNω [33] or a combination of cytokines IFNα, IFNβ, and IFNδ to produce active cytokines in the infected cells and alter the replication of co-infected PRRSV [29]. These Type I IFNs were previously shown to have high antiviral activity against PRRSV, and when inserted into a PRRSV p129 infectious clone viral replication was reduced by 70–90% [18,29]. We hypothesized that expressing Type I IFNs with high antiviral activity in conjunction with PRRSV replication could rescue the suppressed innate immune responses and potentially boost adaptive immunity for increased PRRSV protection.

In the current study, we have demonstrated in two separate experiments that our replication-competent MLV PRRSV vaccine candidate that expresses three separate Type I IFNs (IFNα, IFNβ, and IFNδ) with different antiviral activities and signaling pathways provides vaccination responses and protection against subsequent challenge with virulent PRRSV similarly to the two commercial PRRSV MLV vaccines.

This study has several limitations. First, the vaccinated pigs were challenged with NADC34, a strain heterologous to the two commercial vaccines and our IFNmix vaccine candidate. Second, the vaccine strains were previously attenuated, while our IFNmix construct was being attenuated by the expression of the interferons and separating the impact of the IFNs on immunity versus attenuation in our current dataset is difficult. This shows the potential of the IFNmix construct as a vaccine candidate in that it is as effective as commercial vaccines against NADC34 but using a virulent backbone. With this reverse genetics platform, additional virulent strains could potentially be attenuated for use as vaccines. Third, although both experiments showed similar trends, the overall effect was lower in the second experiment. This could have been a result of different genetic backgrounds between the two cohorts of pigs, environmental differences, or storage conditions.

## 5. Conclusions

Our study evaluated a novel replication-competent PRRSV MLV candidate, IFNmix, engineered to express three distinct Type I interferons (IFNα, IFNδ, IFNω) to counteract PRRSV-induced immune suppression. Across two independent experiments, IFNmix demonstrated comparable immune responses and protection against PRRSV challenge relative to two commercial MLV vaccines. While the study confirms the feasibility of cytokine-expressing PRRSV MLV vaccines, further research is needed to optimize attenuation strategies and evaluate cross-protection against diverse PRRSV strains. The findings highlight the potential of reverse genetics to enhance PRRSV vaccine design by integrating antiviral cytokines to improve immune responses. Key findings are highlighted:IFNmix induced similar anti-PRRSV antibody levels and immune responses as commercial MLV vaccines.IFNmix vaccination resulted in comparable or lower PRRSV replication post-challenge relative to commercial MLVs.Serum cytokine analysis indicated that IFNmix modulated immune responses similarly to commercial vaccines.Lung lesion severity was not significantly different between IFNmix and commercial MLV-vaccinated groups.Differences between experimental results underscore the need for further optimization and validation.IFNmix’s use of a virulent PRRSV backbone suggests the potential for further attenuation-based vaccine improvements.

## Figures and Tables

**Figure 1 vaccines-13-00413-f001:**
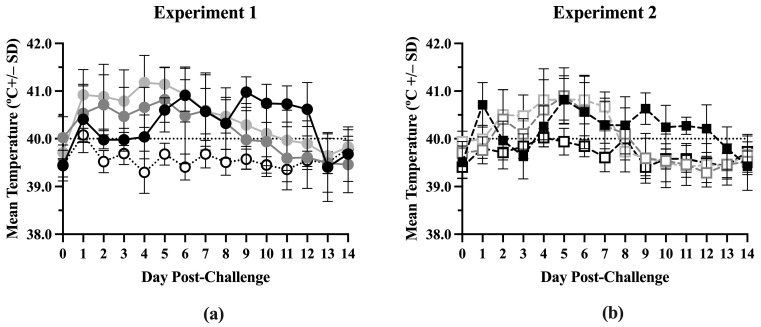
Mean temperature by treatment post-challenge. Body temperatures were recorded daily post-challenge for all treatment groups in both experiments. Data points shown are mean body temperature (in °C) +/– standard deviation. Values above the black dotted line are considered to be fever. (**a**) Experiment 1. Black line/circles = Sham/PRRSV; dark gray line/circles = IFNmix/PRRSV; light gray line/circles = ComMLV1/PRRSV; dotted line/white circles = Sham/Sham. (**b**) Experiment 2. Dashed black line/black square = Sham/PRRSV; dark gray dashed line/open square = IFNmix/PRRSV; light gray dashed line/open square = ComMLV2/PRRSV; black dashed line/open square = Sham/Sham.

**Figure 2 vaccines-13-00413-f002:**
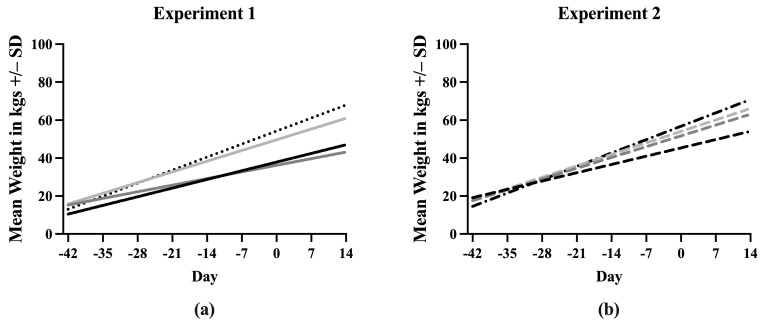
Simple linear regression of weight in kgs by treatment group. Pigs were weighed weekly throughout the course of the studies. Weights in each treatment group were averaged and simple linear regression was used to fit a line to the data points. The slope of each line was compared with those of the other treatments in GraphPad Prism 10.0. (**a**) Experiment 1. Black line = Sham/PRRSV; dark gray line = IFNmix/PRRSV; light gray line = ComMLV1/PRRSV; black dotted line = Sham/Sham. (**b**) Experiment 2. Black dashed line = Sham/PRRSV; dark gray dashed line = IFNmix/PRRSV; light gray dashed line = ComMLV2/PRRSV; black dot/dash line = Sham/Sham.

**Figure 3 vaccines-13-00413-f003:**
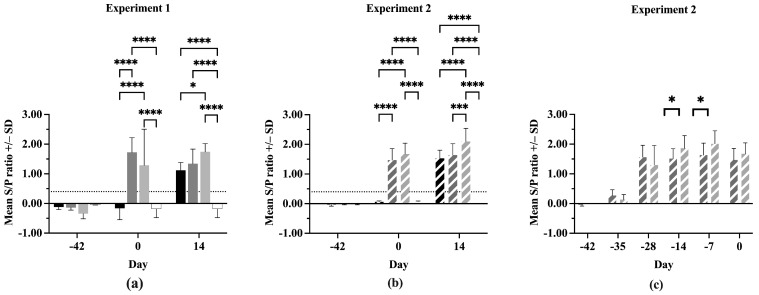
Antibody response of pigs to PRRSV challenge. The serum concentration of anti-PRRSV in all pigs was determined using the IDEXX PRRS X3 Ab test ELISA before vaccination, before PRRSV challenge, and 14 days post-challenge. Values shown are mean S/P ratio +/– standard deviation. Differences between treatments at each time point were tested using a two-way ANOVA with Tukey’s multiple comparisons test (**a**) Experiment 1. Black = Sham/PRRSV; dark gray = IFNmix/PRRSV; light gray = ComMLV1/PRRSV; white = Sham/Sham. (**b**) Experiment 2. Black diagonal = Sham/PRRSV; dark gray diagonal = IFNmix/PRRSV; light gray diagonal = ComMLV2/PRRSV; white with black diagonal = Sham/Sham. (**c**) Experiment 2 showing the development of anti-PRRSV serum antibodies after vaccination with IFNmix MLV and ComMLV2 until PRRSV challenge. Dark gray diagonal = IFNmix; light gray diagonal = ComMLV2/PRRSV. * *p* < 0.05, *** *p* < 0.001, **** *p* ≤ 0.0001.

**Figure 4 vaccines-13-00413-f004:**
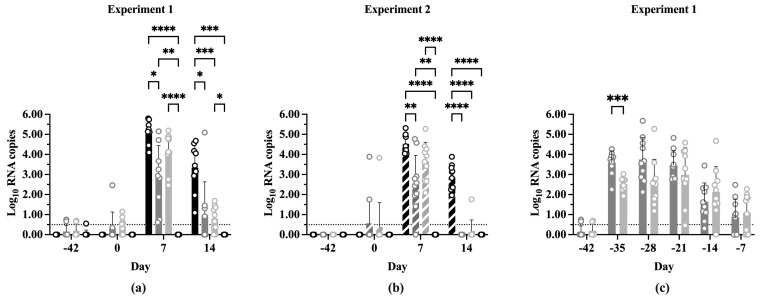
Log_10_ PRRSV RNA copies in serum pre- and post-challenge. PRRSV replication was determined by RT-qPCR for all treatment groups vaccination, before PRRSV challenge, and 7 and 14 dpc. RNA copy number was calculated from CT by comparison with a standard curve included on each plate. Values shown are mean log_10_ RNA copy number +/− standard deviation. Differences between treatments at each time point were tested using a two-way ANOVA with Tukey’s multiple comparisons test. The black dotted line indicates the limit of detection of the RT-qPCR assay. Open circles indicate values for individual samples. (**a**) Experiment 1. Black = Sham/PRRSV; dark gray = IFNmix/PRRSV; light gray = ComMLV1/PRRSV; white = Sham/Sham. (**b**) Experiment 2. Black diagonal = Sham/PRRSV; dark gray diagonal = IFNmix/PRRSV; light gray diagonal = ComMLV2/PRRSV; white diagonal = Sham/Sham. (**c**) Experiment 1, showing the replication profile of IFNmix MLV and ComMLV2 from vaccination until PRRSV challenge. Dark gray = IFNmix/PRRSV; light gray = ComMLV2/PRRSV. * *p* < 0.05, ** *p* < 0.01, *** *p* < 0.001, **** *p* ≤ 0.0001.

**Figure 5 vaccines-13-00413-f005:**
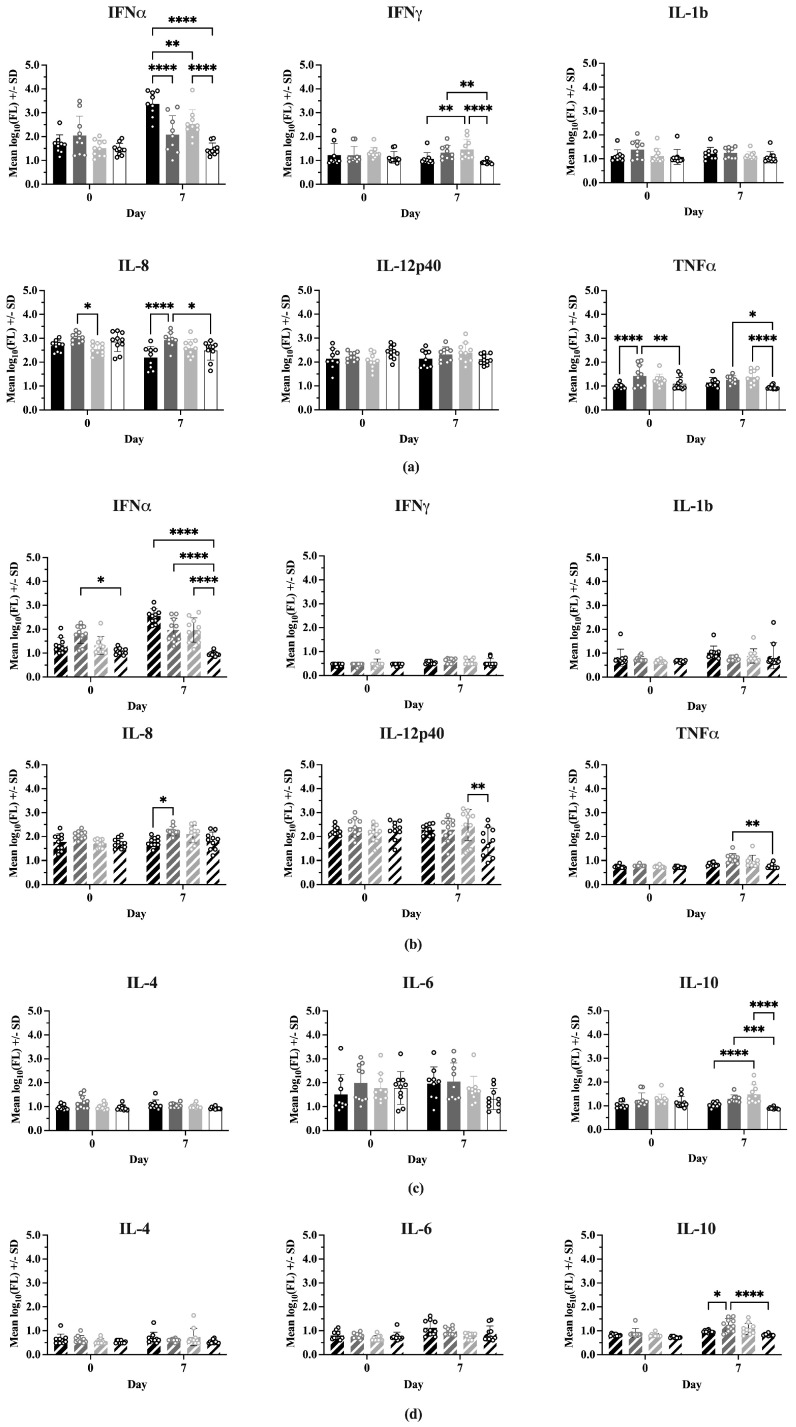
Cytokine levels were measured using the ProcartaPlex Porcine Cytokine Panel run on a Luminex MAGPIX instrument. Median fluorescence values were log_10_ transformed prior to analysis by a two-way ANOVA. Open circles indicate values for individual samples. (**a**) Experiment 1—Th1-associated cytokines; IFNα, IFNγ, IL-1B, IL-8, TNFα, IL-12p40. Black = Sham/PRRSV; dark gray = IFNmix/PRRSV; light gray = ComMLV1/PRRSV; white = Sham/Sham. Comparison between treatments at Day 0 and 7 days post-challenge (dpc). (**b**) Experiment 2—Th1-associated cytokines. Black diagonal = Sham/PRRSV; dark gray diagonal = IFNmix/PRRSV; light gray diagonal = ComMLV2/PRRSV; white diagonal = Sham/Sham. Comparison between treatments at Day 0 and 7 dpc. (**c**) Experiment 1—Th2-associated cytokines; IL-4, IL-6, IL-10. Black = Sham/PRRSV; dark gray = IFNmix/PRRSV; light gray = ComMLV1/PRRSV; white = Sham/Sham. Comparison between treatments at Day 0 and 7 dpc. (**d**) Experiment 2—Th2-associated cytokines. Black diagonal = Sham/PRRSV; dark gray diagonal = IFNmix/PRRSV; light gray diagonal = ComMLV2/PRRSV; white diagonal = Sham/Sham. Comparison between treatments at Day 0 and 7 dpc. Statistical significance between treatment groups is indicated with asterisks (*): * *p* < 0.05, ** *p* < 0.01, *** *p* < 0.001, **** *p* ≤ 0.0001.

**Figure 6 vaccines-13-00413-f006:**
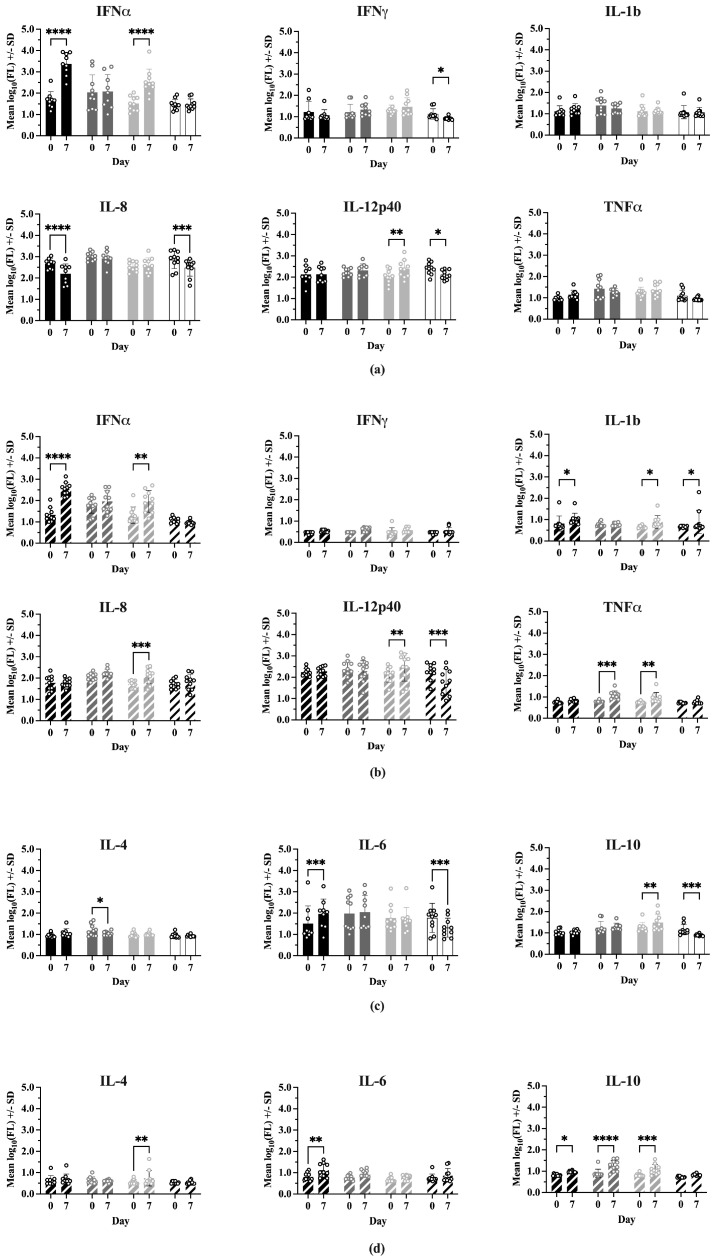
Cytokine levels were measured using the ProcartaPlex Porcine Cytokine Panel run on a Luminex MAGPIX instrument. Median fluorescence values were log_10_ transformed prior to analysis by a two-way ANOVA. Open circles indicate values for individual samples. (**a**) Experiment 1—Th1-associated cytokines; IFNα, IFNγ, IL-1B, IL-8, TNFα, IL-12p40. Black = Sham/PRRSV; dark gray = IFNmix/PRRSV; light gray = ComMLV1/PRRSV; white = Sham/Sham. Comparison between Day 0 and 7 dpc within treatment (**b**) Experiment 2—Th1-associated cytokines. Black diagonal = Sham/PRRSV; dark gray diagonal = IFNmix/PRRSV; light gray diagonal = ComMLV2/PRRSV; white diagonal = Sham/Sham. Comparison between Day 0 and 7 dpc within treatment. (**c**) Experiment 1—Th2-associated cytokines; IL-4, IL-6, IL-10. Black = Sham/PRRSV; dark gray = IFNmix/PRRSV; light gray = ComMLV1/PRRSV; white = Sham/Sham. Comparison between Day 0 and 7 dpc within treatment. (**d**) Experiment 2—Th2-associated cytokines. Black diagonal = Sham/PRRSV; dark gray diagonal = IFNmix/PRRSV; light gray diagonal = ComMLV2/PRRSV; white diagonal = Sham/Sham. Comparison between Day 0 and 7 dpc within treatment. Statistical significance between treatment groups is indicated with asterisks (*): * *p* < 0.05, ** *p* < 0.01, *** *p* < 0.001, **** *p* ≤ 0.0001.

**Figure 7 vaccines-13-00413-f007:**
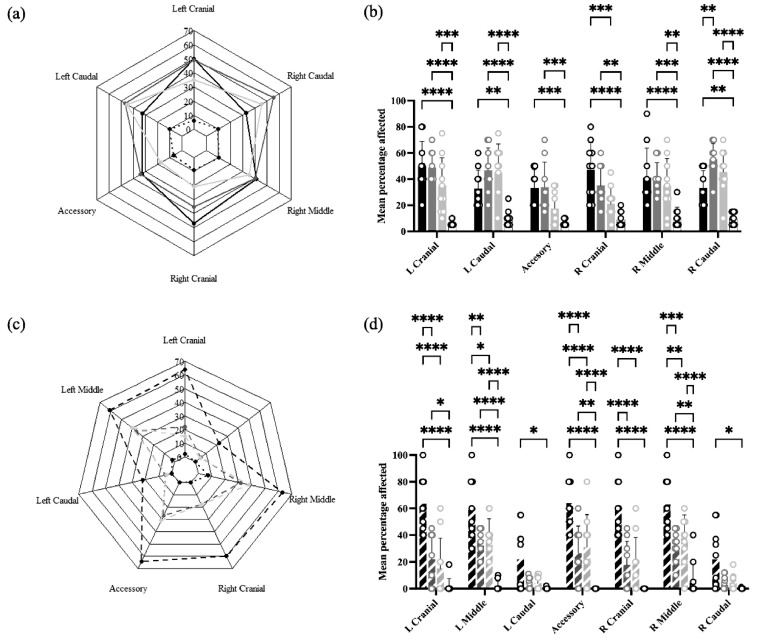
Macroscopic lung lesions were scored at necropsy on 14 dpc. Mean lesion scores by lung lobe for each treatment were plotted as radar graphs, and lesion scores between treatments for individual lobes were analyzed by a two-way ANOVA. Columns represent the mean percentage of lobe affected and open circles represent the values for individual pigs. (**a**) Experiment 1—Radar graph of mean lesion scores by lung lobe for each treatment. Black line = Sham/PRRSV; dark gray line = IFNmix/PRRSV; light gray line = ComMLV1/PRRSV; black dotted line = Sham/Sham. (**b**) Experiment 1—Comparison between treatments of the mean percentage of lobe affected for individual lung lobes. Color scheme is the same as in (**a**). (**c**) Experiment 2—Radar graph of mean lesion scores by lung lobe for each treatment. Black dashed line = Sham/PRRSV; dark gray dashed line = IFNmix/PRRSV; light gray dashed line = ComMLV2/PRRSV; black dotted line = Sham/Sham (**d**) Experiment 2—Comparison between treatments of the mean percentage of lobe affected for individual lung lobes Color scheme is the same as in (**c**). Statistical significance between treatment groups is indicated with asterisks (*): * *p* < 0.05, ** *p* < 0.01, *** *p* < 0.001, **** *p* ≤ 0.0001.

## Data Availability

The data supporting the findings of the study are available within the Appendix A of this article.

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
