# Peer review of "Vaccine Efficacy of a Replication-Competent Interferon-Expressing Porcine Reproductive and Respiratory Syndrome (PRRS) Virus Against NADC-34 Challenge"

_vaccines, 2025, doi:10.3390/vaccines13040413_

Round 1

Reviewer 1 Report

Comments and Suggestions for Authors

Porcine reproductive and respiratory syndrome virus (PRRSV) poses significant challenges to swine industry due to its high mutation rates and its ability to suppress innate immune responses. In the manuscript entitled ‘Vaccine Efficacy of a Replication-Competent Interferon-Expressing Porcine Reproductive and Respiratory Syndrome (PRRS) Virus against NADC-34 Challenge’, Miller et al. developed a novel modified live vaccine, IFNmix, by inserting genes of three type I IFN subclasses into a replication-competent PRRSV. They found that IFNmix, by expressing multiple IFNs, can potentially enhance innate and adaptive immune responses, offering a promising approach to improving PRRSV vaccine efficacy. Their findings provide a novel mean to designing PRRSV vaccines in the future.The manuscript is well-organized. I have a few suggestions that may help improve the manuscript.

  1. Ln 44, missing ‘]’ for the  [1-3.
  2. Line 94-95, the first three letters of the restriction enzyme name should be italicized, the rest should be the same.
  3. Line 106-119, only experiment 2 is introduced, so what’s about experiment 1?
  4. Line 116, ‘2ml 1×104 TCID50 /ml’ to ‘2 ml 1×104 TCID50 /ml’.
  5. For figures, compared to different colors, identifying different treatment groups with different hollow or solid symbols is easier for readers to recognize, especially for color blind individuals.

Author Response

Comments 1. Ln 44, missing ‘]’ for the [1-3.

 Response 1. Thank you for catching this error. It has been corrected.

Comments 2. Line 94-95, the first three letters of the restriction enzyme name should be italicized, the rest should be the same.

Response 2. Thank you for catching this error. It has been corrected.

Comments 3. Line 106-119, only experiment 2 is introduced, so what’s about experiment 1?

Response 3. The experimental designs were very similar for experiments 1 and 2. The text has been updated to reflect the similarities and differences.

Comments 4. Line 116, ‘2ml 1×104 TCID50 /ml’ to ‘2 ml 1×104 TCID50 /ml’.

Response 4. Thank you for catching this inconsistency. It has been corrected.

Comments 5. For figures, compared to different colors, identifying different treatment groups with different hollow or solid symbols is easier for readers to recognize, especially for color blind individuals.

Response 5. Thank you for pointing out that the figures may not be accessible to all readers. The figures have been converted to black and white to increase the accessibility to those with sight issues.

Reviewer 2 Report

Comments and Suggestions for Authors

I am glad to have had the opportunity to read this  work. 

The paper is very detailed with lots of abbreviations and significant amounts of detail in the results. This makes it a time consuming and challenging read.  As far as I can see from a quick read is that much or most of the methodology and presentation of results and conclusions are sound.

One question I have is about the vector chosen for making the IFNmix MLV. This was originally the P-129 virus which was highly virulent in pigs (as reported in reference 18). The authours then used a DNA clone of this virus to make their new virus expressing the interferon genes. Great. What I would like to know , however, is what does  infectious virus produced from unaltered pCMV-S-P129 clone do to pigs when it is inoculated into them?  Couldn't this material have been an important control in this work to clearly illustrate the value of the three added interferon genes?

Author Response

Comments 1. One question I have is about the vector chosen for making the IFNmix MLV. This was originally the P-129 virus which was highly virulent in pigs (as reported in reference 18). The authors then used a DNA clone of this virus to make their new virus expressing the interferon genes. Great. What I would like to know, however, is what does infectious virus produced from unaltered pCMV-S-P129 clone do to pigs when it is inoculated into them?  Couldn't this material have been an important control in this work to clearly illustrate the value of the three added interferon genes?

Response 1. Thank you for your thoughtful comment and for highlighting the importance of appropriate controls in evaluating vaccine efficacy. We appreciate your insight regarding the use of the unaltered pCMV-S-P129 clone as a potential comparison in this study.

While the original P129 strain was indeed reported as a highly virulent PRRSV2 strain (as noted in reference 28), its virulence is considered moderate when compared to more recent circulating strains, such as the NADC34-like strain, which was used in our study as challenge virus. In this context, our primary goal was to assess the efficacy of the interferon-expressing MLV construct relative to a commercially relevant vaccine—specifically, Ingelvac ATP PRRS MLV (ComMLV1 in Experiment 1 of this study), which was also derived from a P129-like PRRSV2 backbone.

As documented in our previous studies (e.g., Sang et al., 2012; Sang et al., 2014), the replication dynamics and pathogenicity of viruses rescued from the parental P129 clone and those expressing interferon genes have been well characterized in vitro. These studies clearly demonstrated that replication-competent expression of biologically active interferons significantly reduced virus replication and virus-induced cytopathology, providing strong evidence for the antiviral role of the introduced genes.

Although the vaccine manufacturer may have internal data on constructs such as pCMV-S-P129-GFP or pKERMIT, we did not have access to those datasets during the course of this study. Nonetheless, the extensive prior characterization of the P129 strain and its derivatives—along with the use of Inglevac ATP PRRS MLV as a commercially relevant benchmark—provided a robust reference framework for interpreting our results.

In summary, while the unmodified pCMV-S-P129 clone could have served as an additional control, our study design, supported by prior in vitro characterization and the inclusion of an industry-standard vaccine, was sufficient to demonstrate the added value of interferon gene expression in the MLV construct. We hope this explanation adequately addresses your question.